# The Development of Super-Saturated Rebamipide Eye Drops for Enhanced Solubility, Stability, Patient Compliance, and Bioavailability

**DOI:** 10.3390/pharmaceutics15030950

**Published:** 2023-03-15

**Authors:** Dong-Jin Jang, Jun Hak Lee, Da Hun Kim, Jin-Woo Kim, Tae-Sung Koo, Kwan Hyung Cho

**Affiliations:** 1Department of Bio-Health Technology, College of Biomedical Science, Kangwon National University, Chuncheon 24341, Republic of Korea; 2College of Pharmacy and Inje Institute of Pharmaceutical Sciences and Research, Inje University, Gimhae 50834, Republic of Korea; 3Graduate School of New Drug Discovery and Development, Chungnam National University, Daejeon 34134, Republic of Korea

**Keywords:** rebamipide, super-saturated eye drop, physical stability, ophthalmic bioavailability

## Abstract

The present study aimed to develop clear aqueous rebamipide (REB) eye drops to enhance solubility, stability, patient compliance, and bioavailability. For the preparation of a super-saturated 1.5% REB solution, the pH-modification method using NaOH and a hydrophilic polymer was employed. Low-viscosity hydroxypropyl methylcellulose (HPMC 4.5cp) was selected and worked efficiently to suppress REB precipitation at 40 °C for 16 days. The additionally optimized eye drops formulation (F18 and F19) using aminocaproic acid and D-sorbitol as a buffering agent and an osmotic agent, respectively, demonstrated long-term physicochemical stability at 25 °C and 40 °C for 6 months. The hypotonicity (<230 mOsm) for F18 and F19 noticeably extended the stable period, since the pressure causing the REB precipitation was relieved compared to the isotonic. In the rat study, the optimized REB eye drops showed significantly long-lasting pharmacokinetic results, suggesting the possibility of reducing daily administration times and increasing patient compliance (0.50- and 0.83-times lower C_max_ and 2.60- and 3.64-times higher exposure in the cornea and aqueous humor). In conclusion, the formulations suggested in the present study are promising candidates and offer enhanced solubility, stability, patient compliance, and bioavailability.

## 1. Introduction

Rebamipide (REB, 2-(4-chlorobenzoylamino)-3-[2(1H)-quinolinon-4yl]-propionic acid), which enhances the healing of wounded epithelial cell monolayers and the secretion of mucus glycoproteins in gastric mucosal cells, was originally developed and launched as an anti-ulcer drug. In the ophthalmological field, several previous studies demonstrated that REB enhances mucin production in corneal and conjunctival goblet cells [1,2]. In addition, REB was found to attenuate the structural irregularities and haze induced by ultraviolet B radiation in mice, as well as increase the growth of corneal epithelial cells [3,4]. Based on the expansion of drug indications, an REB ophthalmic suspension (Mucosta^®^ ophthalmic suspension UD 2%; Otsuka Pharmaceutical Co., Ltd., Tokyo, Japan) has been clinically used in Japan for the treatment of dry eye and corneal damage since 2012 [5]. It is also gaining attention for moderate or severe dry eye syndrome.

The REB is practically insoluble in water and neutral pH buffers suitable for the ophthalmic administration route [6]. The microparticle suspension is available only in a commercially available dosage form, and there is no clear solution eye drop for clinical use. Before administration in the eye, the suspension should be shaken well for redispersion, because the suspension can form flocculation during storage [7]. A patient administered this milky white suspension of high REB concentration (2%) would suffer from blurred vision shortly after the drop is placed in the eye [8]. In addition, producing a sterile pharmaceutical suspension product is a technically challenging task. The suspension cannot be sterilized by a simple sterile filtration pore size of 0.2 µm, which complicates the manufacturing processes and makes it more expensive [9]. Compared with clear solutions for eye drops, suspensions generally have several disadvantages, including manufacturing, quality control, and user compliance.

To date, several technologies to develop REB nano-suspension using the liposome, bead milling, or high-speed dispersion method have been proposed [10,11,12]. However, as these nanoparticles are part of the nano-suspension systems, there have been some obstacles such as failure of filtration and the use of dialysis for purification. Therefore, there is a clear need for the development of clear solution eye drops for commercial mass production that would enhance patient compliance. REB is part of the carboxylic acid group (pKa = 3.38) in the molecular structure (see Figure 1), and its aqueous solubility is highly dependent on the solution pH [6,13,14]. Accordingly, the pH-modification method can be a promising way to prepare a clear REB aqueous solution. However, for an effective formulation, it is necessary to overcome the challenges of preserving the physicochemical stability of the super-saturated solution.

In the present study, REB was solubilized to a very high concentration using the pH-modification method, and various hydrophilic polymers were evaluated to stabilize the super-saturated clear REB eye drop solution. The excipients for the buffering agent and osmotic agent were investigated in terms of solubility, stability, reasonable manufacturing process, and quality control. In the rat study, the bioavailability of the various ophthalmic tissue was also studied.

## 2. Materials and Methods

### 2.1. Materials

REB was purchased from EstechPharma Co., Ltd. (Hwaseong-si, Gyeonggi-do, Republic of Korea). The commercial product (Mucosta^®^ opthalmic suspension UD 2%) was purchased from Otsuka Pharmaceutical Co., Ltd. (Tokyo, Japan). Polyvinylpurrolidone K30 (PVP K30) was provided by BASF (Ludwigshafen, Germany). Hydroxypropylmethylcellulose 4.5cp (HPMC 4.5cp), hydroxypropylmethylcellulose 15cp (HPMC 15cp), and hydroxypropylmethylcellulose 100cp (HPMC 100cp) were provided by Shin-etsu Chemical Co., Ltd. (Tokyo, Japan). Carbopol 934 was provided by Lubrizol Corporation (Cleveland, OH, USA). Hyaluronic acid and sodium alginate were purchased from Sigma Aldrich Inc. (St. Louis, MO, USA). Carboxymethylcellulose sodium (CMC Na) were provided Ashland Inc. (Covington, OH, USA). D-sorbitol and dextrose was provided by Roquette (Lestrem, France). Aminocarproic acid was purchased from WonPoong Pharm. Co,. Ltd. (Hwaseong-si, Gyeonggi-do, Republic of Korea). L-lysine hydrochloride was provided by Ajinnomoto Co., Inc. (Tokyo, Japan). All other chemicals were of reagent grade and used without further purification.

### 2.2. REB Solubility on pH According to NaOH Concentration

The solubility of REB was evaluated in various concentrations of NaOH solutions. NaOH solutions were prepared by gradually diluting 1N NaOH with water to obtain a 10^−1^~10^−7^ N NaOH concentration (pH 6.13~pH 12.58). An excess amount of REB powder was added to each NaOH solution. Then, the suspended solution was stirred with magnetic barr stirring for 12 h. The suspension was centrifuged at 17,000 rpm for 30 min (LZ-1730R, LABOGENE, Seoul, Republic of Korea), and the supernatant was diluted with the mobile phase used in the high-performance liquid chromatography (HPLC) condition (Section 2.3). The amount of REB was analyzed in triplicate by the HPLC method. The pH measurement for NaOH solutions before and after dissolving REB powder was conducted using a pH-meter (Mettler Toledo S210, Greifensee, Switzerland).

### 2.3. HPLC Condition

The HPLC analysis of REB in the samples was conducted using a Waters 2695 system (Waters, Milford, MA, USA) equipped with a UV-Vis detector (Waters 2487, Waters, Milford, MA, USA). REB was separated by a reverse-phase column (C18, 5 um, 4.6 mm × 150 mm) (Osaka Soda, Osaka, Japan). The mobile phase was a mixture of a phosphate buffer (14.70 mM KH_2_PO_4_/0.41 mM Na_2_HPO_4_, pH 6.45) and acetonitrile (75:25, *v*/*v*). The HPLC analysis was conducted with a flow rate of 0.8 mL/min. The injected volume of the sample was 20 μL, and the UV detection was monitored at 254 nm. Data acquisition and processing were performed using the Waters LC Solution software (Empower 2.0 version).

### 2.4. REB Precipitation pH in Hydrophilic Polymer Solution

The hydrophilic polymer 0.5 g (HPMC 100cp, HPMC 15cp, HPMC 4.5cp, PVP K30, CMC Na, Carbopol 934, Hyaluronic acid, Alginate Na) was dissolved in 90 mL of water with magnetic barr stirring and simultaneous pH measurement, and 1.5 g of REB was suspended in the solution. Then, 5 mL of 0.1 N NaOH solution was added to dissolve the REB completely, and the pH solution was around pH 12~pH 13. The REB and hydrophilic polymer solution was titrated dropwise by 0.1 N HCl solution with pH measurement and magnetic barr stirring to determine the pH at which precipitation occurred. The experiment was performed in triplicate, and precipitation pH was observed visually.

### 2.5. Hydrophilic Polymer on the Suppression of REB Precipitation

A total of 1.5 g of REB and 0.5 g of each polymer for F1~F5 as shown in Table 1 were added to 90 mL of water. Then, 5 mL of 0.1 N NaOH solution was added with magnetic barr stirring and simultaneous pH measurement to adjust to pH 10 or more to dissolve REB and polymers. This solution was titrated to pH 7.4 dropwise by 0.1 N HCl solution and diluted with water to a total of 100 mL. Next, 15 mL of the solution was transferred to a 20 mL vial and stored in a 40 °C chamber (HB-501M, Hanbaek, Bucheon, Republic of Korea). REB precipitation was visually observed every day during storage. The experiment was performed in triplicate.

### 2.6. Buffering Agents on the Suppression of REB Precipitation

A total of 1.5 g of REB and 0.5 g of HPMC 4.5cp for F6~F10, as shown in Table 2, were added to 90 mL of water. Then, 4.2 mL of 1 N NaOH solution was added with magnetic barr stirring and simultaneous pH measurement to adjust to pH 10~pH 11. When REB and HPMC 4.5cp were completely dissolved, each amount of the buffering agent and glycerin as an osmotic agent were added and dissolved. This solution was titrated to pH 7.2~pH 7.6 dropwise by a 0.1 N HCl solution and diluted with water to a total of 100 mL. The solution was then moved into a controlled clean bench and filtered with a syringe filter (0.2 μm pore size, Millex^®^-GV, Merck Millipore Korea Ltd., Seoul, Republic of Korea) for sterilization and transferred to a pre-sterilized vial. The prepared eye drops were stored in a 40 °C chamber (HB-501M, Hanbaek, Bucheon, Republic of Korea). REB precipitation was visually observed every day during storage. The experiment was performed in triplicate.

### 2.7. Osmotic Agent and Osmolarity on the Suppression of REB Precipitation

A total of 1.5 g or 2.0 g of REB, 0.5 g of HPMC 4.5cp, and 0.3 g of aminocaproic acid for F11~F16, as shown in Table 3, were added to 90 mL of water. Then, 4.2 mL of 1 N NaOH solution was added with magnetic bar stirring and simultaneous pH measurement to adjust to pH 10~pH 11. When REB, HPMC 4.5cp, and aminocaproic acid were completely dissolved, each amount of osmotic agent was added and dissolved. This solution was titrated to pH 7.2~pH7.6 dropwise by 0.1 N HCl solution and diluted with water to a total of 100 mL. The solution was then moved into a controlled clean bench and filtered with a syringe filter (0.2 μm pore size, Millex^®^-GV, Merck Millipore Ltd.) for sterilization and transferred to a sterilized vial. The prepared eye drops were stored in a 40 °C chamber (HB-501M, Hanbaek, Bucheon, Republic of Korea). REB precipitation was visually observed every week during storage. The experiment was performed in triplicate.

### 2.8. Na Analysis from Representative Precipitation Formulation Using LIBS (Laser-Induced Breakdown Spectrometer)

A representative unstable formulation (F12) showed heavy precipitation at 40 °C after storage for 20 weeks, and the sample of 10 mL was centrifuged at 15,000 rpm for 10 min (LZ-1730R, LABOGENE, Seoul, Republic of Korea). Then, the supernatant was removed by a pipette and the spin-down precipitation was transferred to a plate and dried at room temperature for 24 h. This dried sample was analyzed for the quantitative determination of Na by Tandem LIBS experiments (J200 tandem LIBS system, Applied Spectra Inc., West Sacramento, CA, USA), equipped with a frequency quadrupled Nd:YAG laser operating at a wavelength of 266 nm. Element-specific radiation emitted from the laser-induced plasma was collected by optical fibers and analyzed using a Czerny-Turner-type spectrometer with 6-channel charge-coupled device (CCD) detection involving a gate delay of 1 μs and a gate width of 1.05 ms. The spectral analysis was performed between 190 and 1050 nm (0.1 nm in spectral resolution) for every laser shot. The Axiom software from the manufacturer of the instrument was used for the collection of the LIBS data. The National Institute of Standards and Technology (NIST) 610 was used for the certified reference material (trace elements in glass) containing Na of a specific 9.94% (*w*/*w*). The content of Na in the sample was determined by comparing the intensity of Na emission peaks between the sample and NIST 610 [15].

### 2.9. Long-Term Stability of Optimized Formulation

Samples of F14 and F15 were prepared on a 100 mL scale according to the method in Section 2.7. Afterwards, 5 mL of samples were packed in each sterilized vial and stored in chambers at 25 °C and 40 °C, and the appearance, pH, and REB content were analyzed at 0 and 24 weeks. The appearance of the samples was visually observed. The pH was measured by a pH meter (Mettler Toledo S210, Greifensee, Switzerland). To measure the REB content, the REB eye drops were diluted with 50% ethanol to be REB 150 μg/mL and filtered through a syringe filter (pore size of 0.45 μm) and analyzed according to the above HPLC conditions.

### 2.10. Eye Distribution in Rats

Animal studies were approved in advance by the Institutional Animal Care and Use Committee of Chungnam National University (202103A-CNU-053; Daejeon, Republic of Korea). Then, 7-week-old Sprague Dawley males (weight: 180–220 g), provided by Orient Bio (Seongnam, Republic of Korea), were used. Before the experiment, the animals were acclimatized in a constant temperature and humidity animal room for 1 week. 

Twenty μL of 1.5% REB (F14) and commercial product were instilled into the rat’s left eye under isoflurane anesthesia. Four rats were used at each time point. The left eyes were excised after euthanasia at 0.083, 0.25, 0.5, 1, 3, and 7 h after administration. The aqueous humor, cornea, vitreous, and retina were sequentially separated from the collected eyes. After adding 200 μL of distilled water to each tissue, the tissues were homogenized using a sonic dismembrator (FB50; Fisher Scientific, Pleasanton, CA, USA) for 30 s.

The Agilent 1100 HPLC system (Agilent Technologies, Santa Clara, CA, USA) coupled to an API 4000 triple quadrupole mass spectrometer (AB Sciex, Framingham, MA, USA) was used for the quantification of F14 and commercial product in ophthalmic tissue. Protein precipitation was induced by adding 20 µL of the internal standard (IS, terfenadine) solution and 160 µL of acetonitrile to 20 µL of tissue homogenate. Then, the sample was suspended using a vortex mixer and centrifuged at 13,500 rpm. Next, 150 µL of the supernatant was placed in an analysis vial, and 5 µL was injected into the HPLC-MS/MS system for subsequent analysis. REB and IS were separated via a ZORBAX^®^ Phenyl 3.5 µm, 2.1 × 50 mm column using 0.1% formic acid in the DW/ACN 50:50 isocratic elution method. In the ESI multiple reaction monitoring mode, REB and terfenadine were monitored at *m*/*z* 371.2 to 216.0 and 472.3 to 436.4. The quantification range was 10 to 10,000 ng/mL and had linearity and specificity within the range that met the FDA bio-analytical method guidance [16].

When calculating the PK parameter, a series of concentration values, according to the sampling order, were regarded as individual values. The maximum concentration (C_max_) and time of C_max_ (T_max_) were determined directly from each concentration–time profile. The area under the curve was calculated using the linear trapezoidal formula as described previously [17].

## 3. Results and Discussion

### 3.1. REB Solubility on pH According to NaOH Concentration

As shown in Figure 2, aqueous REB solubility was highly dependent on the pH according to NaOH concentration. The saturated solubility of REB sharply increased at more than 0.01 N NaOH (initial pH 11.81), and a REB solubility of 22.51 mg/mL was achieved at 0.1 N NaOH (initial pH 12.58). In the pH measurement, the initial pH more than 12 at each concentration of >0.01 N NaOH shifted to a lower pH 7 after the solubilization of REB. REB was highly solubilized with the consumption of NaOH since the carboxylic acid group in the structure was deprotonated and ionized due to the reaction of acid-base neutralization [12]. The significant pH shift from the initial to the lower REB solution pH supported the acid-base neutralization reaction between REB and NaOH, and the addition of NaOH and increase in pH was a very effective way to dissolve REB to a high REB concentration of >2%. Although REB is practically insoluble in water (<0.03 mg/mL), NaOH dissolved REB to a 700-fold higher concentration with a simple pH-modification [18].

### 3.2. REB Precipitation pH in Hydrophilic Polymer Solution

The precipitation pH of REB was determined with the dropwise addition of the 1 N HCl solution to 1.5% REB and 0.5% hydrophilic polymer solution (see Figure 3). The hydrophilic polymers such as HPMC 100cp, HPMC 15cp, HPMC 4.5cp, PVP K30, and CMC Na commonly showed a precipitation pH of <7.0. However, the control water solution without any hydrophilic polymer resulted in a precipitation pH of 7.73. The ionic polymers including Carbopol 934, hyaluronic acid, and alginate Na showed a precipitation pH of >8 and promoted the precipitation of REB more compared to the control water solution. Precipitation pH from ionic polymers was thought to be the result of complex factors, including pKa of carboxyl acid group, chemical structure, and molecular size. It was difficult to correlate the precipitation pH to any specific single property of the polymer. However, the difference in precipitation pH among them would be due to the velocity of forming precipitation. The precipitation pH was observed instantly after the addition of 1 N HCl solution. The cellulose derivatives (HPMC and CMC Na) and PVP K30 were very useful to maintain a super-saturated REB solution at an acceptable physiologically of around pH 7–pH 8 [19,20]. These polymers suppressed the REB precipitation at 0.5% concentration, since they have a high affinity with REB and inhibit the bonding between the REB molecules in the solution [21,22]. All three HPMCs physically stabilized REB solution regardless of the molecular weight and viscosity of the polymer. Therefore, the hydrophilic polymers such as HPMCs, PVP K30 and CMC Na were selected for further formulation study with the stabilizing super-saturated REB solution.

### 3.3. Hydrophilic Polymer on the Suppression of REB Precipitation

The preliminary formulations of F1–F5 containing hydrophilic polymers were prepared with a target 1.5% REB concentration and a physiologically acceptable pH 7.4 for the determination of REB precipitation occurrence days. As shown in Figure 4, the REB concentration was much higher than the solubility of REB at a neutral pH (0.013% *w*/*v*) and indicated that the formulations were in a super-saturated solution state [23]. Thus, stabilizing the solution without REB precipitation was a critical quality issue in the development of the REB formulation. In the results, F1–F3 (HPMCs) showed a longer precipitation occurrence period compared to F4 (PVP K30) and F5 (CMC Na). Comparing the three viscosity HPMCs, it was found that F3 (HPMC 4.5cp) with low viscosity maintained longer physical stability, with an average number of precipitation occurrence days of 16 compared to the higher viscosity of F1 (HPMC 100cp) and F2 (HPMC 15cp). In the results, HPMC, a non-ionic polymer, was confirmed to be more effective in the suppression of REB precipitation than PVP K30 and CMC Na [24]. HPMC 4.5cp of low viscosity was selected for further formulation study in terms of precipitation suppression and was found to be more suitable for sterile filtration with its slight resistance to pressure and manufacturing handling than the higher viscosity HPMC 100cp and 15cp [25].

### 3.4. Buffering Agents on the Suppression of REB Precipitation

The pH of the REB formulation was critical for solubilization and stabilization without the occurrence of REB precipitation. Therefore, to maintain the target pH 7.4, various buffering agents were tested with the pharmaceutically acceptable formulation (F6–F10) in the use of safe inactive ingredients (see Figure 5). F7, F9, and F10 containing boric acid, aminocaproic acid, and L-lysine, as a buffering agent, respectively, showed a longer average number of precipitation occurrence days of ≥19 compared to F6 and F8. Among these three buffering agents, aminocaproic acid from F9 made the most stable formulation with the occurrence days of 26 with HPMC 4.5cp. Na^+^ from buffering agents such as Na_2_B_4_O_7_·10H_2_O and Na_2_HPO_4_ would force REB to precipitate through the salting out reaction with the carboxylic acid group in the REB structure [26]. However, aminocaproic acid suppressed REB precipitation more compared to a single use of HPMC 4.5cp. Therefore, aminocaproic acid was selected as a buffering agent in the further osmotic agent evaluation.

### 3.5. Osmotic Agent and Osmolarity on the Suppression of REB Precipitation

To achieve the optimized formulation, F11–F16 were tested for the suppression of REB precipitation (see Figure 6). In the osmolarity agents for F11–F13, D-sorbitol suppressed precipitation for 5 weeks, which was much longer than F9, as shown in Figure 5. The selected formulation with HPMC 4.5cp, aminocaproic acid, and D-sorbitol resulted in the synergistic suppression of REB precipitation. Moreover, the lower osmolarity of F14 (230 mOsm) and F15 (180 mOsm) noticeably extended the weeks of precipitation occurrence by >20. The osmolarity for commercially available eye drops in dry eye treatment was in the wide range of 145~407 mOsm, as reported by Tong et al. [27]. Thus, the osmolarity for F14 and F15 was in the commercially available osmotic range without concern regarding eye irritation. The osmolarity of formulation was critical in the suppression of REB precipitation and acted as a highly significant stressor for the precipitation of REB. When comparing the REB concentrations of F13 (1.5% & 295 mOsm) and F16 (2.0% & 290 mOsm), F13 was found to keep clear solution for a longer number of weeks (5 weeks versus 4 weeks). Thus, the target REB concentration was confirmed to be 1.5% or less.

The visual appearance of F11–F15 at 20 weeks is shown in Figure 7. F14 and F15 maintained a clear solution for over 20 weeks. However, F11, F12, and F13 showed REB precipitation within 10 weeks. The degree of REB precipitation was severe in the order of F12 ≫ F13 > F11. F12, containing NaCl as an ionic osmotic agent, promoted REB precipitation more compared to both F13 and F11, which contained dextrose and D-sorbitol as a non-ionic osmotic agent, respectively [26]. Na^+^ resulted in REB precipitation with salt formation. In F14 and F15, clear solutions were observed due to the reduction in formulation osmolarity. Accordingly, F14 and F15 were selected as optimized formulations in terms of their physical stability and ability to suppress REB precipitation.

### 3.6. Na Analysis from the Representative Precipitation Formulation Using LIBS

Element analysis of the representative severe precipitation from F12 at the time point of 20 weeks was performed using the LIBS equipment. NIST 610 was used as a standard reference to identify and approximately assay the Na element in the sample. Specific emission peaks from the Na element at 588–591 nm were compared between the precipitation sample from F12 and NIST 610 (Figure 8). The results of this analysis revealed that when compared with the sensitivity of the emission peak of the NIST 610, Na was identified with the overlapped peaks of Na, and the concentration of Na in the samples was approximately similar to the NIST 610 (about 9.94 wt % Na). Although the peak intensity of the sample was lower than that of the standard, similar to the standard reference, the real concentration of Na in the samples would be approximately 9–10% when considering the measurement according to the difference in its physical state (powder state from sample versus glass state from NIST). The powder state was generally underestimated to be a lower % in the peak intensity compared to the glass state of NIST 610 [28]. Therefore, it strongly suggested that REB precipitation is the result of salting out during storage.

### 3.7. Long-Term Stability of Optimized Formulation 

The long-term stability test for F14 and F15 optimized in the suppression of REB precipitation was performed at 25 °C and 40 °C with the major quality attributes of appearance, pH, and REB content (Table 4). The preparation of long-term stability samples was suitable for application in mass production, including all the unit operations and main sterile filtration. In appearance, there was no precipitation for all the tested samples from F14 and F15, and the solution remained clear. Any contamination such as biofilm or microbial growth was not observed, even without preservatives, and the sterile condition was maintained. In the pH measurement, there was a slight decrease (<0.3) compared to the initial pH, but it fell into the physiologically acceptable safe range of pH 6.5~pH 8.0 [19,20]. The contents of REB were higher than 97% at both 25 °C and 40 °C during the 6-month period, indicating that there was no significant change in the drug contents. REB has a stable chemical moiety in the structure and was stabilized in F14 and F15, demonstrating good stability without any significant chemical degradation. In summary, the results showed that these two formulations were physicochemically and microbiologically stable for at least 6 months at 25 °C and 40 °C and ensured long-term stability.

### 3.8. Eye Distribution in Rats

Figure 9 shows the concentration profile of REB over time in ocular tissues following the administration of F14 and commercial products. Table 5 shows the PK parameters. In both administration groups, the drug concentrations were in the following order: cornea, aqueous humor, retina, and vitreous humor. The commercial product group quickly reached the highest concentration at 0.25 h in all ocular tissues and then rapidly disappeared. However, the optimized REB eye drop (F14) reached the highest concentration slowly at 0.375 to 1.75 h compared to the existing commercial formulation, showing a relatively low peak concentration, but a long-lasting pattern. In the aqueous humor and cornea, which were the main tissues of interest in the present study, the C_max_ values of the F14 group were 27,055 and 39,556 ng/g, respectively, i.e., 0.50 and 0.83 times lower than those of the commercial product group. However, the AUC values were 87,844 and 154,892 ng·h/g, i.e., 2.60 and 3.64 times higher than those of the commercial product group. The sustained profile of the F14 formulation was expected to reduce side effects and increase the duration of the drug compared to conventional formulations, thus resulting in a reduction in the number of administrations.

The higher exposure in the F14 group despite a lower concentration of 1.5% compared to the 2% commercial product group was thought to be because F14 was a solution state (Versus a suspension state of commercial product). In general, solutions have been reported to have a higher amount of intraocular permeation compared to suspensions because all the drug in solution is a molecular form that can be absorbed into ocular tissues [29]. HPMC, a hydrophilic polymer used in the F14 formulation, contributed to penetration into ocular tissues by enhancing the residence time of PPD on the mucosal surface. Several studies described that HPMCs increased ocular tissue penetration and retention of drugs [30,31]. In terms of viscosity, given that the viscosity of human tears is 1.5 mPa·s, an eye drop will last around 2–3 min in the eye after the instillation of the drop, which largely limits the bioavailability of the drug to the eye [32]. However, the viscosity of F14 was greater than tear viscosity and increased the residence time in the eye and enhanced bioavailability [20]. Unlike other tissues, retinal exposure was higher in the commercial product group. However, since the retina was not a target site for REB, lower exposure was considered more beneficial in terms of side effects. The prolonged profile of F14 was expected to decrease side effects and increase the duration of PPD compared to conventional formulations, reducing the number of required daily administrations.

## 4. Conclusions

In the present study, the suppression of REB precipitation in the formulation was investigated for the preparation of clear REB 1.5% eye drops. In the solubilization of REB, the pH-modification method using NaOH and HPMC 4.5cp was found to be very successful in terms of reaching a high concentration of REB. The sequential approaches to select aminocaproic acid as a buffering agent and D-sorbitol as an osmotic agent resulted in a remarkable extension of the physically stable period to 5 weeks. Furthermore, hypotonicity in the optimized formulation (F14 and F15) made it possible to show physicochemical long-term stability for 6 months at both 25 °C and 40 °C. In the rat study, F14 prepared with HPMC 4.5cp, safe inactive ingredients, and hypotonicity of 230 mOsm showed long-lasting bioavailability in the ophthalmic tissue compared to the commercial suspension eye drops. In conclusion, the clear eye drop formulation (F14) in the present study was a promising candidate that offered enhanced solubility, stability, patient compliance, and bioavailability.

## Figures and Tables

**Figure 1 pharmaceutics-15-00950-f001:**
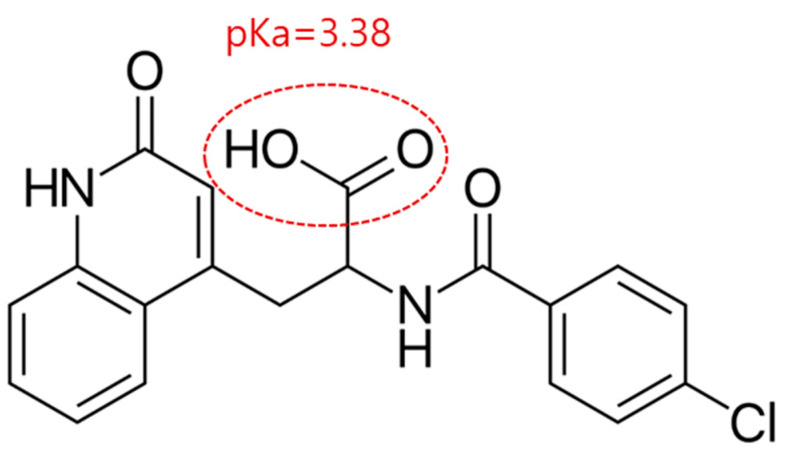
The chemical structure of rebamipide.

**Figure 2 pharmaceutics-15-00950-f002:**
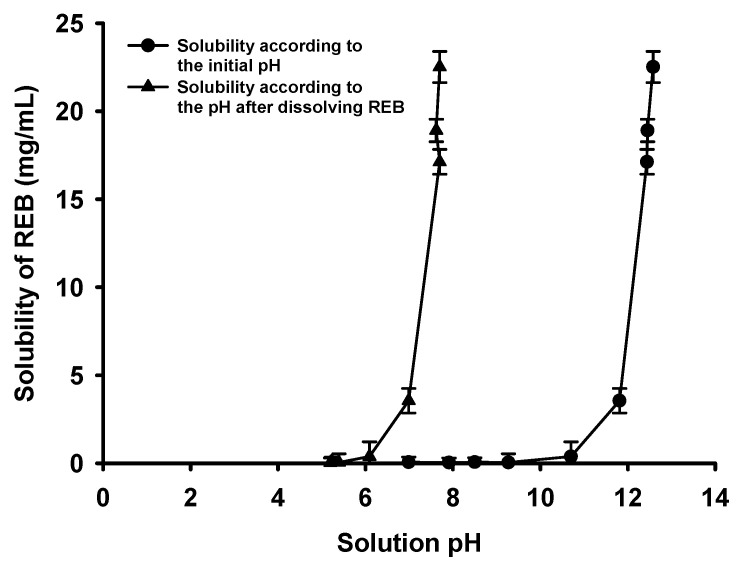
The pH-solubility of REB.

**Figure 3 pharmaceutics-15-00950-f003:**
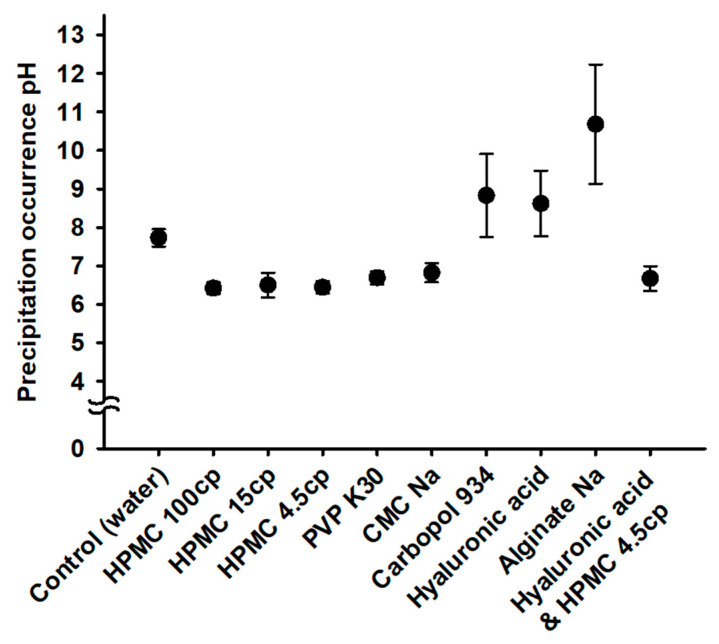
Precipitation pH of REB for each hydrophilic polymer solution.

**Figure 4 pharmaceutics-15-00950-f004:**
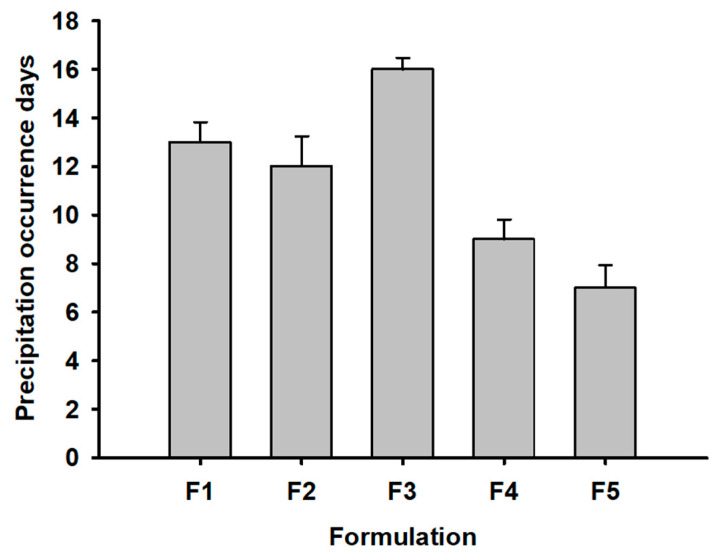
Precipitation occurrence days in preliminary formulations (F1–F5).

**Figure 5 pharmaceutics-15-00950-f005:**
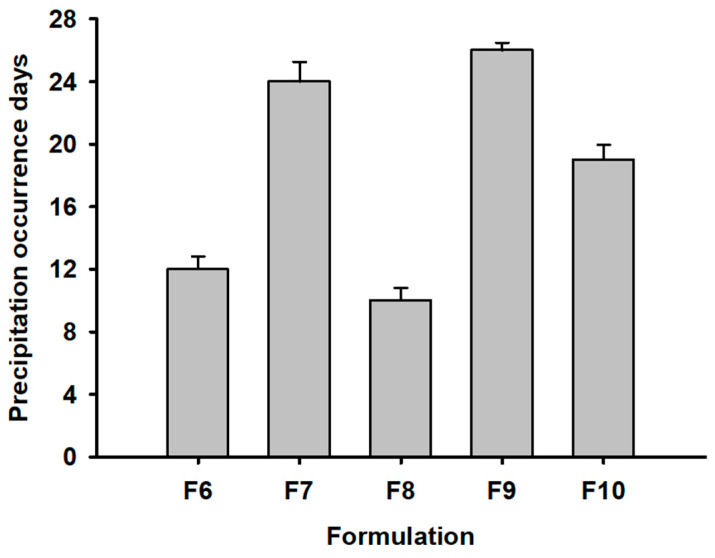
Precipitation occurrence days in REB Formulations (F6–F10).

**Figure 6 pharmaceutics-15-00950-f006:**
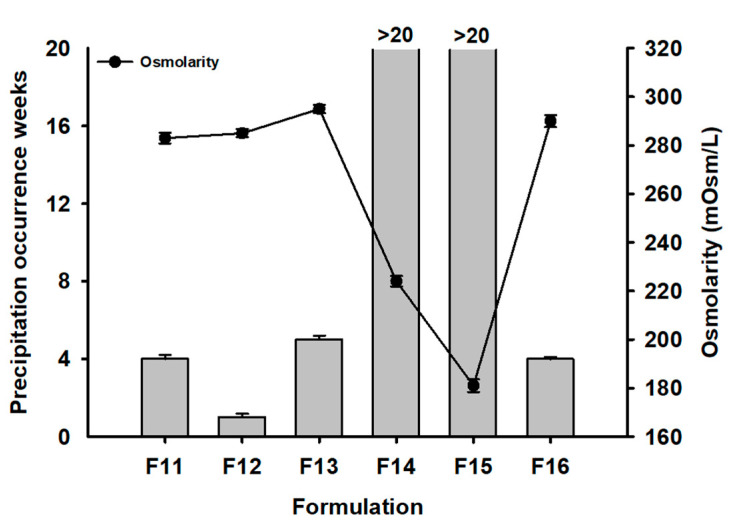
Precipitation occurrence weeks and osmolarity in REB Formulations (F11–F16).

**Figure 7 pharmaceutics-15-00950-f007:**
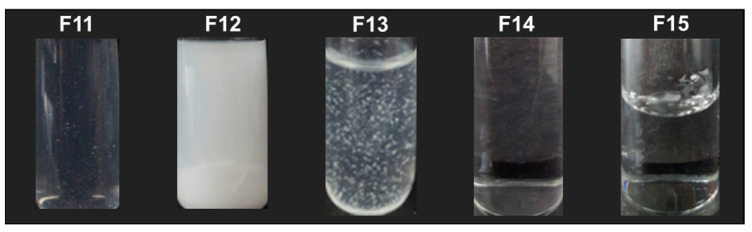
Visual observation of REB Formulations (F11–F15) after long-term stability test.

**Figure 8 pharmaceutics-15-00950-f008:**
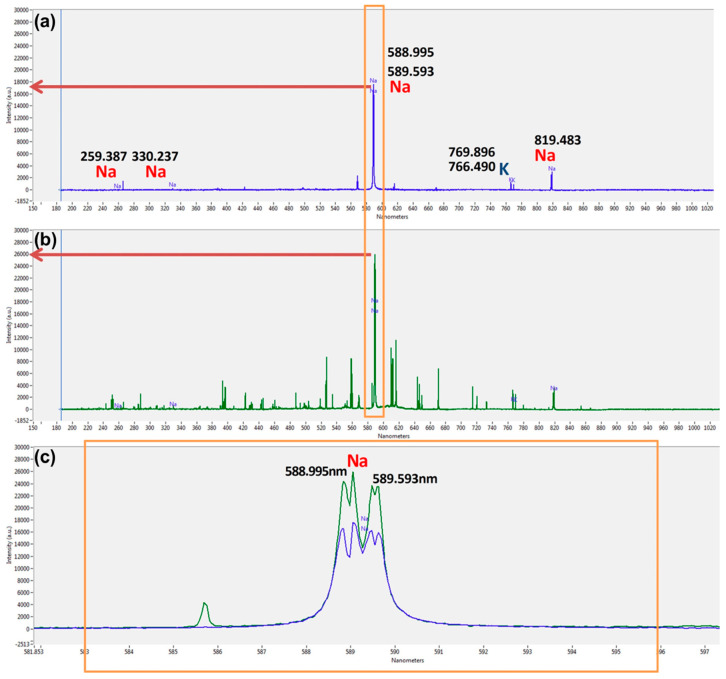
LIBS emission spectrum for (**a**) representative REB precipitation, (**b**) standard reference NIST 610, (**c**) enlarged overlay of emission peaks of Na from REB precipitation and NIST 610.

**Figure 9 pharmaceutics-15-00950-f009:**
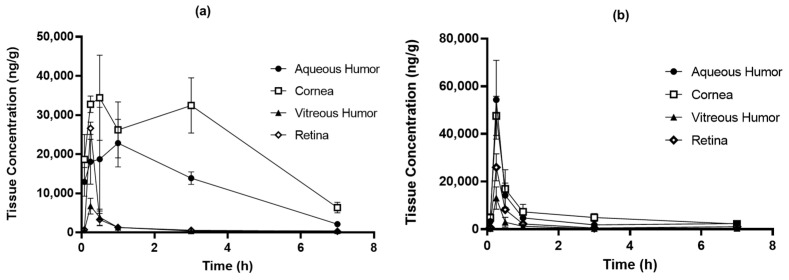
REB ocular concentration profiles after eye drop of (**a**) F14 and (**b**) commercial product in rats (mean ± SD, n = 4).

**Table 1 pharmaceutics-15-00950-t001:** Composition of preliminary formulations (F1–F5).

Formulation	F1	F2	F3	F4	F5
REB (g)	1.5	1.5	1.5	1.5	1.5
1 N NaOH (mL)	5.0	5.0	5.0	5.0	5.0
HPMC 100cp (g)	0.5				
HPMC 15cp (g)		0.5			
HPMC 4.5cp (g)			0.5		
PVP K30 (g)				0.5	
CMC Na (g)					0.5
1 N HCl (mL)	Adjustment to pH 7.4				
Water (mL)	Adjustment to 100 mL				

**Table 2 pharmaceutics-15-00950-t002:** Composition of REB Formulations (F6–F10).

Formulation	F6	F7	F8	F9	F10
REB (g)	1.5	1.5	1.5	1.5	1.5
HPMC 4.5cp (g)	0.5	0.5	0.5	0.5	0.5
1 N NaOH (mL)	4.2	4.2	4.2	4.2	4.2
H_3_BO_3_ (g)	0.31	0.09			
Na_2_B_4_O_7_ • 10H_2_O (g)	0.12				
Na_2_HPO_4_ (g)			0.37		
Aminocaproic acid (g)				0.3	
L-lysine (g)					0.4
Glycerine (g)	1.51	1.89	1.49	1.7	1.56
1 N HCl (mL)	Adjustment to pH 7.2~7.6				
Water (mL)	Adjustment to 100 mL				

**Table 3 pharmaceutics-15-00950-t003:** Composition of REB Formulations (F11–F16).

Formulation	F11	F12	F13	F14	F15	F16
RBP (g)	1.50	1.50	1.50	1.50	1.50	2.00
HPMC 4.5cp (g)	0.50	0.50	0.50	0.50	0.50	0.60
1 N NaOH (mL)	4.20	4.20	4.20	4.20	4.20	5.50
Aminocaproic acid (g)	0.30	0.30	0.30	0.30	0.30	0.30
Dextrose (g)	3.25					
NaCl (g)		0.23				
D-sorbitol (g)			3.55	2.45	1.64	3.00
1 N HCl (mL)	Adjustment to pH 7.2~7.6					
Water (mL)	Adjustment to 100 mL					

**Table 4 pharmaceutics-15-00950-t004:** Results of F14 and F15 in the long-term stability test.

Storage	Quality Attribute	Time Point	F14	F15
25 °C	Appearance	Initial	Clear solution	Clear solution
		24 weeks	Clear solution	Clear solution
	pH	Initial	7.62 ± 0.01	7.49 ± 0.01
		24 weeks	7.45 ± 0.01	7.41 ± 0.01
	Content (%)	Initial	98.19 ± 0.45	97.68 ± 0.71
		24 weeks	97.82 ± 1.12	97.68 ± 2.11
40 °C	Appearance	Initial	Clear solution	Clear solution
		24 weeks	Clear solution	Clear solution
	pH	Initial	7.62 ± 0.01	7.49 ± 0.01
		24 weeks	7.43 ± 0.02	7.42 ± 0.01
	Content (%)	Initial	98.19 ± 0.45	97.68 ± 0.71
		24 weeks	98.22 ± 1.31	98.22 ± 1.76

**Table 5 pharmaceutics-15-00950-t005:** REB PK parameters after eye drop of (a) F14 and (b) commercial product in rats (mean ± SD, n = 4).

PK Parameters	(a)			
	Aqueous Humor	Cornea	Vitreous Humor	Retina
T_max_ (h)	0.875 ± 0.250	1.750 ± 1.443	0.375 ± 0.144	0.313 ± 0.125
C_max_ (ng/g)	27,055 ± 9644	39,556 ± 6347	5728 ± 2621	20,952 ± 11,447
AUC_last_ (ng·h/g)	87,844 ± 13,093	154,892 ± 20,308	7064 ± 2183	9994 ± 1961
PK Parameters	(b)			
	Aqueous Humor	Cornea	Vitreous Humor	Retina
T_max_ (h)	0.250 ± 0.000	0.250 ± 0.000	0.250 ± 0.000	0.250 ± 0.00
C_max_ (ng/g)	54,336 ± 16,561	47,598 ± 8244	13,082 ± 4674	25,973 ± 5656
AUC_last_ (ng·h/g)	33,737 ± 4371	45,731 ± 11,963	7145 ± 1546	15,364 ± 1651

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
