# Peer review of "The Development of Super-Saturated Rebamipide Eye Drops for Enhanced Solubility, Stability, Patient Compliance, and Bioavailability"

_pharmaceutics, 2023, doi:10.3390/pharmaceutics15030950_

Round 1

Reviewer 1 Report

Section 2.10. The drop sizes of the prepared solution and the commercial suspension  are missing.

Information about number of animals and groups are also missing.

I strongly suggest to provide an additional figure to show solubility versus pH as histogram. This will show pH profile solubility curve. 

Figure 2 the units of NaOH concentration is missing from X-axis; so it should read log [NaOH] (concentration units).

The manuscript needs extensive English language revision and proofreading by a native speaker. The manuscript is difficult to follow up and disrupting the follow of the sentence.

The authors mentioned ''The ionic polymers including Carbopol 934, hyaluronic acid, and alginate Na showed a precipitation pH of >8 and promoted the precipitation of REB more compared to the control water solution. Can you provide explanation for that? 

Why hyaluronic acid showed exceptionally precipitation pH lower than the other anionic polymers?

The design of F16 does not show consistency. the authors used higher concentrations of the drug (2%) and high concentration of sorbitol; although you used lower sorbitol concentrations with a drug concentration of 1.5%. This led to early precipitation. Can you justify this? 

NaCl caused salting out not salt formation. Please correct.

One of the weakest point is this. To get correct explanation on the in vivo behavior and pharmacokinetics. The viscosity of the optimized formula and the commercial suspension must be measured. This is because precorneal residence time is an essential factor cannot be overlooked. 

Another important characterization technique is in vitro release studies between the optimized and commercial product  is missing.

Reviewer 2 Report

line 65 Maybe it's better write "chemical structure"

line 93 The first time you use HLPC, you would write it High liquid performance analysis

Reviewer 3 Report

The authors presented the development of rebamipide (REB) containing eye drops, where REB was super-saturated.  A pH- modification method, and various hydrophilic polymers were applied to stabilize the super-saturated REB solution. The excipients such as the buffering agent and osmotic agent were investigated in terms of solubility, stability. An in vivo animal experiment was used to evaluate the bioavailability of the formulation in the various ophthalmic tissue.

The article is logically structured, but some errors and shortcomings can be mentioned:

1. Please provide the ingredients of the eye drops used as a reference and compare possible differences in bioavailability depending on this.

2. What was the amount of eye drops used in the animal experiments?

3. In section 3.7, the microbiological cleanliness of the preparation is also analyzed, but no acceptable tests were performed for this, visual observation is not sufficient. Please delete these parts or perform appropriate microbiological tests.

Reviewer 4 Report

The authors developed the clear aqueous rebamipide (REB) eye drops for enhanced solubility, stability, patient compliance, and bioavailability. However, they didn't proceed the safety evaluation, especially for the irritation test. There are some issues should be addressed.

1. Line 184, the left eyes were excised after euthanasia at 0.083, 0.25, 0.5, 1, 3 and 7 hours after administration. Please explain the reason for the setting of timepoints.

2. Is the pharmacological activity of REB affected due to the reaction of acid-base neutralization between the REB and NaOH? 

3. Line248, the authors mentioned that the formulations were in a super-saturated solution state. I don't think the statement is reasonable. Actually, the compound in the "super-saturated solution" is mainly a type of salt formed during the process of the reaction of REB and NaOH. 

4. The osmolarity for the F14 and F15 is lower than that of the normal tear fluid, which probably results in irritation and should be supported by the  irritative assess.

Round 2

Reviewer 1 Report

Minor comments:

Section 2.4 HPLC condition should come before section 2.3 for better flow of the readability of the manuscript.

Page 5 line 186 .20 μL . Do not start the sentence with number. Change the number to letter Twenty μL.

In the conclusion section you mentioned F14 is promising. You need to summarize the main features of the formulation-related characterstics that led to this decision.

Author Response

Thank you for your nice comments. We revised all the things on your comments. Please see the revised manuscript.